# Applying citizen science to engage families affected by ovarian cancer in developing genetic service outreach strategies

Colleen M. McBride[1]*, Gavin P. Campbell[1], Jingsong Zhao[1], Rebecca D. Pentz[2], Cam Escoffery[1], Michael Komonos[3], Kelly Cannova[4], Janice L. B. Byrne[5], Nancy M. Paris[6], James R. Shepperd[7], Yue Guan[1]

1 Department of Behavioral, Social and Health Education Sciences, Emory University, Atlanta, GA, United States of America, 2 Department of Hematology and Oncology, Emory School of Medicine, Atlanta, GA, United States of America, 3 Emory School of Medicine, Atlanta, GA, United States of America, 4 The OVERRUN Ovarian Cancer Foundation, Overland Park, KS, United States of America, 5 Department of Obstetrics and Gynecology, University of Utah Health Sciences Center, Salt Lake City, UT, United States of America, 6 Georgia Center for Oncology Research & Education, Atlanta, GA, United States of America, 7 Department of Psychology, University of Florida, Gainesville, FL, United States of America

* cmmcbri@emory.edu

**Data Availability Statement:** The Emory Institutional Review Board has restricted the public sharing of the data. The underlying the results presented in the study are available from Mr.

## Abstract

Citizen science (CS) approaches involving non-professional researchers (citizens) as research collaborators has been used infrequently in health promotion generally and specifically, in cancer prevention. Standardized CS approaches may be especially useful for developing communication interventions to encourage families to consider cancer genetic services. We engaged survivors of ovarian cancer and their close relatives as CS collaborators to collect and help interpret data to inform content for a website, printed invitation materials, and short-message reminders. We applied an implementation quality framework, and posed four research questions regarding the feasibility of CS: recruitment, data collection, data quality and evaluation of the experience. CS members were recruited through three networks: clinical sites, local and national cancer support organizations, and online ovarian cancer patient support groups. The professional research team operationalized theory-aligned CS tasks, five data collection options, question banks/scripts for creating surveys, structured interviews, online training and ongoing support from research coaches. 14 CS members agreed to the 12-week and 20-hour commitment for an honorarium. CS members opted to do both qualitative and quantitative assessments. CS members collected 261 surveys and 39 structured interviews. The largest number of surveys were collected for Task 1 (n = 102) to assess survivors' reactions to different possible options for motivating survivors to visit a study website; 77% of this data were complete (i.e., no missing values). Data collected for tasks 2, 3, 4, and 5 (e.g., assessment of survivors' and relatives' respective communication preferences) ranged from 10 to 58 surveys (80% to 84% completeness). All data were collected within the specified time frame. CSs reported 17 hours of work on average and regarded the experience positively. Our experience suggests that CS engagement is feasible, can yield comprehensive quantitative and qualitative data, and is achievable in a relatively a short timeline.

James Leonard, Chief Information Officer, Rollins School of Public Health, james.leonard@emory.edu.

**Funding:** The research was funded by a grant from the National Cancer Institute (NCI) of the NIH that has been acknowledged in the text. The following authors received some salary support from the NCI grant: Drs. McBride, Guan, Pentz, Shepperd; Mr. Campbell, Ms. Zhao and Mr. Komonos. The additional authors did not receive salary support from the NCI grant. However, Ms. Cannova and Dr. Byrne served as Citizen Scientists and received an honorarium from the NCI grant for their service. Ms. Paris volunteered her efforts. The NCI had no role in study design, data collection, data analysis, the decision to publish, or preparation of the manuscript.

**Competing interests:** No authors have competing interests.

## Introduction

Patient and public involvement (PPI) is recommended to be an important responsibility of health program planning and implementation globally [1–3]. PPI offers a process of engaging target audiences in a central role as "co-creaters" of intervention content, approaches, and evaluation methods. In turn, this approach can maximize the likelihood that health promotion programs will be relevant, successful, and acceptable. Indeed, PPI approaches have been associated with increased acceptance, adherence, and sustainability of intervention programs in a number of health contexts [4–8].

PPI in research comprises a continuum of approaches ranging from the public having informal and consultative roles to full control over the program development and implementation process [9, 10]. The conundrum for research collaborations involving the public is to maximize the extent of engagement while considering pragmatic concerns, such as time constraints of externally funded research. To this end, we evaluated the feasibility of engaging citizen scientists, individuals from families affected by ovarian cancer, as part of initial planning for the "Your Family Connects" intervention aimed to encourage uptake of genetic services [U01CA240581-02].

Citizen science (CS) approaches involve non-professionals researcher (henceforth, "citizens") as collaborators in research to shape measures, refine methods, collect and interpret data [11]. While engagement of CSs in environmental sciences research has gained popularity [12, 13], CSs have been engaged infrequently in health promotion generally [14] and specifically, to cancer research [15]. In the field of genomics, a recent review of global human genomics research identified just 32 of 96 basic science and clinical initiatives reporting the use of PPI approaches defined as "active involvement in shaping and guiding research" [16] with only two using CS approaches. One used crowdsourcing (via Amazon Turk) to document the large public knowledge gap between high awareness of breast- and low awareness of ovarian cancer [17]. Nunn and colleagues [16] concluded that genomics- and related translation researchers could benefit from standardized methods to apply PPI.

Standardized CS approaches applied in the context of ovarian cancer may be especially useful for developing communication interventions to consider cancer genetic services. Up to 20% of ovarian cancers are linked to inherited genetic mutations in cancer predisposing genes (Hereditary Breast and Ovarian Cancer (HBOC)) [18]. Life-saving prevention options are available to families confirmed via genetic testing to be mutation carriers (e.g., *BRCA1/2*). Yet, uptake of genetic counseling and testing among those diagnosed with ovarian cancer, referred to as survivors, and their blood relatives is low [19]. Using CS approaches, researchers could collaborate with families affected by ovarian cancer to co-create optimal outreach interventions. CS approaches could enable expeditious collection of broader perspectives via support groups, foundations, and personal networks that provide insights with greater credibility to the target audience.

Moreover, concerns about insurance discrimination and privacy can hinder ovarian cancer families' willingness to discuss inherited risk potential with researchers and health professionals. Such concern may be attenuated by involving CS members in conducting data collection. Survivors might be more willing to contact relatives and in turn, relatives more responsive to provide perspectives when approached by a CS member. Additionally, researchers who are passionate about cancer prevention and control could mentor CSs to gain greater appreciation of the value of research participation.

Current clinical practice and ethical guidelines assign individuals at high risk for inherited genetic mutations the responsibility to inform their first- and second-degree blood relatives that they too may be mutation carriers [20]. Low reported awareness of hereditary risks

among relatives suggests that relying solely on a survivor outreach approach may be inadequate [21, 22]. CS engagement could enable deepened understanding of family communication barriers and receiver resistance factors.

With these considerations, we enlisted survivors of ovarian cancer and their close relatives as CS collaborators to collect and help interpret data to inform content for a website, printed invitation materials, and short-message reminders to be tested in a comparative effectiveness trial. We based the procedures for the approach on implementation quality criteria described by Heigl and colleagues [23]. In implementing the CS approach, we posed four research questions: (1) Can ovarian cancer survivors and close relatives be recruited to partner with professional researchers in a defined start-up window?; (2) Can these CS members be trained to address specific data collection tasks and meet specified data collection goals in a 12-week timeline?; (3) What is the quality of the data collected? And, (4) How do the CS members rate the experience?

## Methods

An interdisciplinary team of researchers initiated the research at Emory University. The CS protocol was reviewed and approved by the Emory University Institutional Review Board [Emory IRB: STUDY00000224]. Here we provide a description of the methods used to engage and collaboration with CS members. The CS implementation was operationalized drawing on a set of minimum criteria for good practice developed by Heigl and colleagues (2020) [Table 1]

**Table 1. "Your family connects" citizen science (CS) methods mapped to quality metrics\*.**

| Quality Criteria | Specific criteria | "Your Family Connects" Implementation |
|---|---|---|
| Scientific standards | State scientific question, hypothesis or goal to be answered with the project | Online Orientation Training |
| | Methods presented in field-specific, appropriate and comprehensible way | Menu of methods with descriptions |
| | New knowledge generated | Online Orientation Training |
| Collaboration | Must be added value for CS and professional researchers | CSs opportunity to shape services for families affected by ovarian cancer. |
| | | Professional researchers opportunity to increase fit of targeted intervention |
| | Objectives are unachievable without the CS collaboration | CS exclusive access to online communities of families affected by ovarian cancer |
| | CS involved in at least one project element (e.g., data collection, data analysis & interpretation) | CS involved in data instrument development, data collection and interpretation |
| | Project definition and objective are open, clear, and communicated in a generally comprehensible manner | Online Orientation Training |
| | Assignment of tasks must be clear and transparent | Online Orientation Training |
| Open Science | Results are published in an open access format | In progress |
| Communication | Different interest groups are addressed accordingly | CS teams include survivors and relatives, representing 4 regions of the U.S. and 9 state |
| | Contact between project management and CSs possible at all times | Coaches available via text and email at all times |
| | CSs receive feedback on the progress and the results of the project | Final meeting of full group of CS to discuss findings; offered co-authorship on manuscripts |
| Ethics | Informed consent is obtained from CSs | One-on-one online conferencing, review of detailed consent form, return signed consent |
| | | Research team used all data collected and expressed sincere appreciation of the CSs work |
| Data management | Prior to data collection, all projects must have established a data management plan | A detailed data analysis plan and its application to intervention development was specified as part of the approved research. |

\*Adapted from Heigl, Kieslinger, Paul, et al., *Citizen Science*: *Theory and Practice*, 2020 [23].

[23]. Six criteria are outlined: scientific standards, collaboration, open science, communication, ethics, and data management, along with specific criterion for each.

## Developing a theory-based intervention

The "Your Family Connects" intervention was conceived to align with the "Foot-in-the-Door" technique (FITD), a stepped approach we wanted to test to encourage families at high risk for ovarian cancer to seek genetic counseling services [24]. The central premise of FITD is that people are more likely to agree to a large request (e.g., providing contact information for close relatives) if they have first agreed to a small, easy request (e.g., visit a website designed exclusively for survivors of ovarian cancer) [25, 26]. The FITD technique may be effective for a variety of reasons including the possibility that it elicits consistency concerns [27] or because of self-perception processes [28] whereby people infer their value from their initial behavior [25]. While the logic of this approach is straight-forward and testable, the focus and content can take many forms that may be more or less appropriate depending on the situation. To date, no one has engaged CSs to operationalize behavioral theory (e.g., what constitutes a small- or large ask).

Ensuring the quality of CS-collected data is critical to the reliability and usefulness of the information. CS members have varying levels of expertise that are likely not comparable to professional researchers [29]. To this end, the professional research team operationalized CS tasks in accordance with scientific standards to follow the FITD theoretical logic: *Task 1*: Identify connected steps of contact that would motivate ovarian cancer survivors (OCS) to seek new information via a website; *Task 2*: Obtain OCSs' perspectives on contacting their blood relatives regarding inherited risk; *Task 3*: Obtain close relatives' perspectives on being contacted by the OCS or professionals; *Task 4*: Collect perspectives on information sharing within families after an ovarian cancer diagnosis; and *Task 5*: Characterize OCS- and close relatives' needs regarding genetic counseling and testing.

## CS toolkit

The research team operationalized five data collection methods for the CS members to choose from: survey, structured interview, personal story, online panel, and role plays options [Table 2]. Additionally, the research team developed a bank of survey items that aligned with each of the tasks and methods including structured interview templates with scripting and open-ended questions [See S1 File for the Citizen Science Toolkit].

## CS recruitment

We initiated recruitment in September 2020. Eligible CS members were 25 years or older, spoke English, had used video-conferencing, and indicated having personal or professional survivorship connections amenable to support data collection. Although initially designed to recruit ovarian cancer survivors and their relatives living in Georgia, the onset of the COVID-19 pandemic required us to shift from an in-person approach to a virtual format. We broadened recruitment to the continental United States. CS members were recruited through three networks: clinical connections made in partnership with community organizations, local and national cancer support organizations, and online ovarian cancer patient support groups [Fig 1]. Additionally, snowball sampling or chain referral sampling was used such that individuals interested in participating were encouraged to refer others they knew who met the eligibility criteria [30].

In network 1, we informed 36 gynecologic oncologists identified by the Georgia Center for Oncology Research and Education (CORE) about the study. Interested oncologists received

**Table 2. Menu of data collection methods and explanation.**

| Methods | Explanations |
|---|---|
| Survey | For this activity, you would identify a group to complete an anonymous short survey. This activity could be taken on by multiple CS participants who would do fewer surveys (10 or so) within survivors (or family members) or a single CS participant could identify a large online listserv community for administering the survey. |
| Structured Interview | For this activity, you would identify individuals who you would interview by phone or via electronic options using a set of open-ended questions. You would be asked to keep the interviews anonymous to the researchers. Each interview would take 30–45 minutes and responses to the questions would be audio-recorded and later coded. |
| Personal story | For this activity, you would identify pairs of individuals (a survivor paired with a close relative) and ask them to tell their story of their relationship since the cancer diagnosis. You would be provided with probing questions to guide the story telling. The stories would be audio-recorded, later coded, and remain anonymous to the researchers. |
| Online panel | For this activity, you would identify 3–5 individual (all survivors or close relatives or a mix of the two) to participate in an online discussion via zoom or another electronic platform. You would pose a set of questions to the panel and the discussion would be audio-recorded and later coded. Panelists would be not be identified in any materials. |
| Role plays | For this activity, you would try out tools that we are developing with survivors [and relatives] and record their comments and reactions to the tool. |

the IRB-approved study flyer and distributed it to patients. For network 2, the study team asked seven local ovarian cancer foundations (e.g. Lilies of the Valley at Alabama) and two national ovarian cancer organizations (Ovarian Cancer Research Alliance; Facing Hereditary Cancer Empowered) to share study flyers with their members. In network 3, the study team received approval to post study flyers on open online patient support groups and national support and discussion groups (n = 5; e.g., American Cancer Society Network Discussion Forum). The flyer directed interested people to complete an on-line eligibility screening questionnaire.

## CS implementation

**Informed consent.** Once deemed to be eligible, a study investigator interviewed the prospective CSs via Zoom or phone call. The study investigator provided the participant with an overview of the study, tasks to be implemented, time commitment and data collection expectations (20 hours over 12 weeks), compensation ($550), and their rights as a CS. CSs were then asked to review a detailed written consent document and to recontact study investigators if they remained interested. Those willing to participate returned the electronic consent form via encrypted email at Emory.

**Orientation training.** Due to COVID restrictions, we consented and trained the CS members virtually. In advance of the training session, we provided CS members with a 30-minute pre-recorded orientation presentation that described the background rationale for the five selected tasks, and descriptions of each of the five method options. CS members selected a task, data collection method, and whether to work individually or as a team.

**Coaching sessions.** One of two trained research coaches (GC, JZ) worked with each of five teams in several sessions. In session 1, the coaches introduced themselves and discussed their role with the CS team. Each team received data collection goals (e.g., 50 surveys per CS member; 2–3 structured interviews per CS member). In session 2, CS members discussed how they would engage with their personal or professional networks to collect data. The coaches also guided a discussion of the team's perceptions of what was feasible with respect to data collection so they could settle on their data collection methods and sources. Coaches also gave brief tutorials on how to use Webex to conduct structured interviews and Qualtrics for surveys. In Session 3, teams finalized the surveys and interview guides, and discussed network

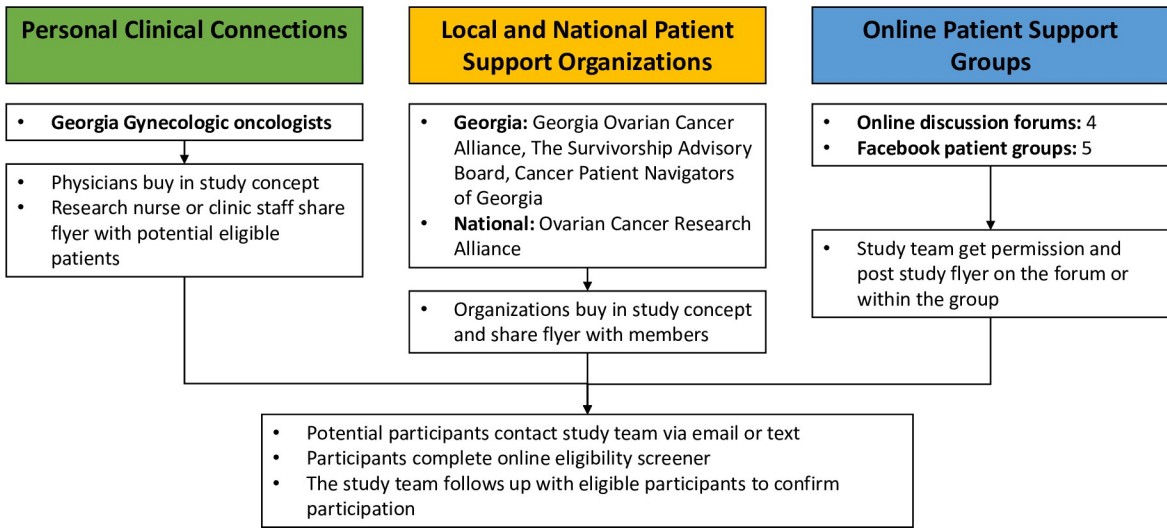

**Fig 1. Recruitment strategies.**

outreach strategies. They selected questions from a bank of questions we developed, making edits to questions and interview guides as needed. Most edits were directed at increasing clarity and personalizing the introduction they would use with their networks.

Coaches kept track of each citizen scientists' progress, reminding them of the study deadline and their progress toward their goals. Coaches also initiated Webex structured interviews so that we could save recordings to the study team's account. Prior to scheduled interviews, coaches sent copies of the specific interview guides and Webex instructions to the CS member. CS members who reported concern about reaching their goal consulted with the coaches to find more sources of interviews or organizations to share their survey links.

Team boundaries loosened considerably as the process of data collection progressed. For example, coaches made the CS members' in one team aware of other team's challenges with data collection. When Team 2 ran into difficulties linking into professional networks, Team 3 offered to include Team 2's survey within Team 3's broader networks. This process of collaboration was facilitated by the brevity and distinctness of the surveys.

### Statistical analysis

Descriptive statistics were used to compute counts and indicators of distribution including means, standard deviations and proportions for quantitative and qualitative data collected for each task.

## Results

### Recruitment of ovarian cancer survivors and close relatives

Over the course of six weeks of CS recruitment (Sept-October, 2020), 22 individuals reached out to the study team to consider serving as a citizen scientist, 21 completed the screener, 4 were deemed ineligible (never used video chat; no professional/personal networks). The majority of prospective CS members reported hearing about the opportunity through ovarian cancer support groups (n = 10). Of the 17 deemed eligible, 2 declined after discussing the study with a co-investigator and 1 was lost to follow-up; 14 chose to proceed as CS members. All CS members were female, 11 were survivors and 3 were close relatives of someone who had

experienced ovarian cancer, 3 lived in the Southwestern U.S., 3 lived in the Midwest, 1 in the Northeast, and 7 were in the Southeast. The members represented six foundations.

## Data completeness

Each team opted to do both qualitative and quantitative assessments. No team opted for personal stories, online panels, or role plays. In total, CS members collected 261 surveys and 39 structured interviews [Table 3]. A summary of data collected in the designated period (by January 22, 2021) appears in Table 3 by task. Survey data were collected primarily via the CS members' personal and professional listservs, that is electronic mailing list software applications used by U.S. based ovarian cancer advocacy groups. Team 1 collected the largest number of surveys (Task 1: n = 102) assessing survivors' reactions to different possible options for motivating survivors to visit the study website; 77% of the data were complete (i.e., no missing values). Teams 2, 3, 4, and 5 collected between 10 and 58 surveys (80% to 84% completeness). All survey data was collected within the specified time frame.

Of the 39 structured interviews completed, 13 were completed for Task 1, 4 for Task 2, 7 for Task 3, 7 for Task 4, and 8 for Task 5 [Table 3]. Completeness of structured interviews was based on adherence to the interview script (i.e., asked all questions on the script). Across structured interviews, 50–82% were complete. However, the interviews varied considerably in length (Mean = 26 mins (SD = 16.3 minutes; Range: 7 to 78 minutes). Additionally, five scheduled interviews (1 survivors; 4 relatives) were not recorded at the participant's request. Instead, the CS interviewer took notes. Seven interviews were conducted in writing because the CS member was undergoing treatment. In these cases, the duration of the interview was unknown.

To supplement areas where the data was most sparse (Task 5), the study team hosted two panels that the CS members could voluntarily attend in week 10 and week 11 of the data collection window. A member of the research team conducted the panel discussions, prompting the CS members to answer either from their own experiences or from the experiences expressed by participants interviewed over the preceding weeks. In the final week, the study team reported the study results to CS members in a virtual session.

## CS members' rating of the experience

CS members completed a survey evaluating their experience. On average CS members reported spending 16.8 hours (SD = 8.1) on the project, less on average than the estimated 20-hour commitment [Table 4]. Ratings of experience with training, assistance from coaches, and fellow team members fell in the middle of the five-point where 1 indicated "less than

**Table 3. Citizen science collected data and completeness.**

| Citizen Science Task | Surveys (N) | Minimum Survey Completeness[a] (%) | Structured Interviews (N) | Minimum Structured Interview Completeness[a] (%) |
|---|---|---|---|---|
| Task 1: Identify connected contact steps to motivate survivors to see new information via a website | 102 | 77 | 13 | 82 |
| Task 2: Obtain survivor perspective on contacting relatives | 45 | 84 | 4 | 50 |
| Task 3: Obtain relative perspective on being contacted | 46 | 83 | 7 | 75 |
| Task 4: Collect perspectives on information sharing within families after an ovarian cancer diagnosis | 58 | 81 | 7 | 100 |
| Task 5: Characterize survivors' and blood relatives' information needs regarding genetic counseling & testing | 10 | 80 | 8 | 80 |

[a] % Completeness = Item with lowest completion/total number of items included.

**Table 4. Citizen scientists' evaluations of their experience.**

| Domains | Mean | SD |
|---|---|---|
| Time spent on project activities (hours) | 16.8 | 8.06 |
| Satisfaction with: | | |
| Activity you chose | 4.6 | 0.65 |
| Group/Team Interactions | 4.4 | 0.93 |
| Compensation received | 4.9 | 0.53 |
| Overall experience | 4.6 | 0.85 |
| *(1-Very Dissatisfied– 5-Very Satisfied)* | | |
| The amount of time I devoted to this project took: *1-Much less time than I expected– 5-Much more time than I expected* | 2.6 | 0.63 |
| What I learned from this project was: *1- Much less than I expected– 5- more than I expected* | 3.4 | 0.87 |
| The tasks I was asked to do were: *1-Much easier than I expected– 5-Much harder than I expected* | 3.1 | 0.73 |
| The training I received for this project | | |
| *1-Much less than I needed/wanted– 5-Much more than I needed/wanted* | 2.9 | 0.36 |
| The help I received from the coaches | | |
| *1-Much less than I needed/wanted– 5-Much more than I needed/wanted* | 3.0 | 0.00 |
| The assistance I received from my team members | | |
| *1-Much less than I needed/wanted– 5-Much more than I needed/wanted* | 2.8 | 0.60 |
| Participating in this project enabled me to: | | |
| Make new connections within the ovarian cancer survivorship community | 4.0 | 1.04 |
| Learn more about the needs of ovarian cancer survivors and their families | 3.9 | 0.95 |
| Gain insights about the experiences of ovarian cancer survivors that will be useful to me | 3.6 | 1.15 |
| Make a valuable contribution to the community of ovarian cancer survivors | 4.3 | 0.83 |
| *(1-Strongly disagree– 5-Strongly agree)* | | |
| Experience with: | | |
| Research Training | 3.9 | 0.77 |
| Research Team | 4.3 | 0.61 |
| Coach | 4.4 | 0.74 |
| *(1-Poor– 5-Excellent)* | | |

expected or needed", 3 indicated "as expected" and 5- "more than expected/needed". Satisfaction ratings averaged in the middle of the scale for time devoted the project, what they learned, and the difficulty of the tasks, but toward the top of the scale for overall experience.

CS members also averaged above the scale midpoint in their ratings of how useful the experience was for making new connections within the ovarian cancer survivorship community, learning more about the needs of ovarian cancer survivors and their families, gaining insights about the experiences of ovarian cancer survivors that will be useful to me, and making valuable contribution to the community of ovarian cancer survivors.

## Discussion

Using the Heigl et al., framework, we implemented a successful effort to recruit a diverse group of CS members representing ovarian cancer survivors and their close relatives. Fourteen people participated from four regions of the U.S. CS members worked with the guidance of a research coach who convened meetings via online conferencing, provided technical support related to data collection and provided updates on progress towards their goals.

Our results show that in a 12-week timeframe we showed favorable outcome, process and feasibility indicators aligned with CS evaluation frameworks [31]. These include CSs collected more than 200 surveys, and completed 39 structured interviews, far more data than our

research team could have accomplished using lower order PPI (e.g., ad hoc focus groups). The CS teams' ready access to large online support groups dedicated to ovarian cancer survivors and their families and the ongoing technical support they received from research coaches facilitated their success. Our success compares favorably to most published program development research, where formative data collection often results in small samples (<30; e.g. [32]). Additionally, CSs reported that they were able to accomplish these goals by committing an average of 17 hours over the 12 weeks, slightly below what we had asked them to anticipate.

Relating to the scientific process and outcome indicators [31], the data collected were complete overall and comprehensively covered four of the five assigned tasks. Study investigators led two debriefing panel discussions to gauge CS members' perspectives regarding the broader information needs related to ovarian cancer that they heard expressed or that they felt were important for the message intervention we were developing.

Relating to outcomes and impact [31], CSs rated their experiences aligned well with their expectations. They viewed the coaching and meetings to be helpful—not too little or too much. They reported feeling as though they had made a contribution to the community of ovarian cancer survivors and family, a quality criterion also noted by Heigl's framework. Additionally, CS members left wanting additional opportunities to take on similar roles in other research projects.

Despite these positive outcomes, a few areas need further consideration. Our recruitment strategies failed to engage any male relatives to serve as CS members. CS members reported and the researchers agreed, that they could have benefited from more in-depth training in conducting structured interviews. CS members had little experience in using probing techniques, that is, using open-ended questions to promote critical thinking on the part of the interviewee. Thus, several of the structured interviews were brief and functioned more as survey data. This limited skill set may explain why CSs selected only one of the qualitative data collection options. The additional three qualitative methods all involved less structured interactions with participants. This has been reported elsewhere as a challenge for CS approaches [31].

Additionally, CS members encountered participant mistrust regarding the privacy of the information collected. Several of the CS members reported that their family members or other interviewees were not comfortable having their interviews recorded. Professional researchers and coaches provided CS members with additional rationale for recording (e.g., "the information you provide is so important, we don't want to miss anything") and assurances of privacy (e.g., reiterating IRB privacy requirements), but these assurances often did not assuage the concerns. Assisting CSs in garnering trust in genetics-related research likely will require focused training in conveying the nuances of Institutional Review Board requirements.

Practical implications of this report include that guided by an implementation framework CS engagement is feasible, achievable in a relatively a short timeline (even amidst a pandemic), and can provide comprehensive quantitative and qualitative data. Moreover, this approach enables CSs to meaningfully engage in developing interventions for their own communities, something that supports social justice goals and improves the fit of the intervention approach. This report also provides a detailed description of our CS methods and outputs to enable CS partnerships with professional researchers to apply these methods in their own contexts. Efforts to develop health promotion programs to apply genomic discovery may be especially benefitted by PPI approaches such as CS.

## Supporting information

**S1 File. Citizen science toolkit.**
(PDF)

## Acknowledgments

We thank our citizen scientists for their important contributions in their community data collection outreach efforts: Jess BeCraft, Susan Bossert, Jan Byrne, Kelly Cannova, Jade Gibson, Phyllis Gilbert, Nancy Hicks, Aryn Kinney, Cindy McKinnon Deurloo, Regina Parker, Stacy Saravo, Marilyn Slodki, Summer Southern, and Kamilah Staggers. We gratefully acknowledge Georgia Center for Oncology Research and Education (Georgia CORE), Ovarian Cancer Research Alliance (OCRA), Facing Hereditary Cancer Empowered (FORCE), American Cancer Society Cancer Survivors Network, Ovarian Cancer Support Group, Ovarian Cancer Community for their assistance in recruiting citizen scientists. We thank the Intervention Development, Dissemination and Implementation (IDDI) Core Resource of the Winship Cancer Institute (P30CA138292) for assistance in coaching citizen scientists in survey development (Shaheen Rana, Director).

## Author Contributions

**Conceptualization:** Colleen M. McBride, Gavin P. Campbell, Jingsong Zhao, Cam Escoffery, James R. Shepperd, Yue Guan.

**Data curation:** Gavin P. Campbell, Kelly Cannova, Janice L. B. Byrne, Yue Guan.

**Formal analysis:** Colleen M. McBride, Gavin P. Campbell, Jingsong Zhao, Michael Komonos, Kelly Cannova, James R. Shepperd, Yue Guan.

**Funding acquisition:** Colleen M. McBride, Yue Guan.

**Investigation:** Rebecca D. Pentz.

**Methodology:** Colleen M. McBride, Cam Escoffery, Nancy M. Paris, James R. Shepperd.

**Project administration:** Colleen M. McBride.

**Supervision:** Jingsong Zhao, Rebecca D. Pentz, Nancy M. Paris.

**Writing – original draft:** Colleen M. McBride, Gavin P. Campbell.

**Writing – review & editing:** Colleen M. McBride, Rebecca D. Pentz, Cam Escoffery, Michael Komonos, Kelly Cannova, Nancy M. Paris, James R. Shepperd, Yue Guan.

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
