## [Decision Letter · Decision Letter 0]

27 Aug 2021

PONE-D-21-22428

Applying Citizen Science to engage families affected by ovarian cancer in developing genetic service outreach strategies

PLOS ONE

Dear Dr. McBride,

Thank you for submitting your manuscript to PLOS ONE. After careful consideration, we feel that it has merit but does not fully meet PLOS ONE’s publication criteria as it currently stands. Therefore, we invite you to submit a revised version of the manuscript that addresses the points raised during the review process.

Please make sure that you address the minor comments suggested by reviewers, which I am sure will improve the manuscript. 

We look forward to receiving your revised manuscript.

Kind regards,

Manuel Corpas, PhD

Academic Editor

PLOS ONE

Journal Requirements:

2. Please provide additional details regarding participant consent. In the Methods section, please ensure that you have specified (1) whether consent was informed and (2) what type you obtained (for instance, written or verbal). If your study included minors, state whether you obtained consent from parents or guardians. If the need for consent was waived by the ethics committee, please include this information.

3. Please include your tables as part of your main manuscript and remove the individual files. Please note that supplementary tables (should remain/ be uploaded) as separate "supporting information" files.

“The citizen science project was supported by National Cancer Institute grant U01CA240581-02.”

We note that you have provided funding information within the Acknowledgements Section. Please note that funding information should not appear in the Acknowledgments section or other areas of your manuscript. We will only publish funding information present in the Funding Statement section of the online submission form.

“Yes, the work was funded by an NIH research grant that has been acknowledged in the text.”

7. PLOS requires an ORCID iD for the corresponding author in Editorial Manager on papers submitted after December 6th, 2016. Please ensure that you have an ORCID iD and that it is validated in Editorial Manager. To do this, go to ‘Update my Information’ (in the upper left-hand corner of the main menu), and click on the Fetch/Validate link next to the ORCID field. This will take you to the ORCID site and allow you to create a new iD or authenticate a pre-existing iD in Editorial Manager. Please see the following video for instructions on linking an ORCID iD to your Editorial Manager account: https://www.youtube.com/watch?v=_xcclfuvtxQ

Reviewers' comments:

Reviewer's Responses to Questions

**Comments to the Author**

1. Is the manuscript technically sound, and do the data support the conclusions?

Reviewer #1: Yes

Reviewer #2: Yes

2. Has the statistical analysis been performed appropriately and rigorously? 

Reviewer #1: Yes

Reviewer #2: I Don't Know

3. Have the authors made all data underlying the findings in their manuscript fully available?

Reviewer #1: Yes

Reviewer #2: Yes

4. Is the manuscript presented in an intelligible fashion and written in standard English?

Reviewer #1: Yes

Reviewer #2: Yes

5. Review Comments to the Author

Reviewer #1: The authors aim to evaluate the feasibility of a Citizen Science quality framework, recruiting survivors of ovarian cancer and their close relatives, suggesting that CS engagement is fesible. The manuscript is well written but requries some minor revisions in order to be more comprehensive. Please find my comments as follows:

Introduction

Page 3 line 69, there is no reference for the sentence

“In the initial 63 planning year of a four-year NCI-funded study entitled “Your Family Connects” we applied citizen 64 science to generate intervention content for a message-based outreach intervention to encourage 65 families affected by ovarian cancer to consider genetic service options.”

Page 4, line 75. There is no reference for the sentence “Nunn and colleagues concluded that genomics- and 75 related translation researchers could benefit from standardized methods to apply PPI”. This regards the ref 16, which should be reported also in this sentence, otherwise it seems that is the continuum of ref 17.

Page 4, line 77-78, the sentence “Up to 20% of ovarian 78 cancers are linked to inherited genetic mutations in cancer predisposing genes”, should not be underlined.

Page 4, line 89 It is not clear the transition to concerns, i would suggest to add a sentence on the general concerns before describing each of them in details.

Page 5, line 107-108, i would suggest to include Table 1 in the Methods section and leave in the introduction the reference.

The introduction is quite long and should be shortened, with a focus on Citizens sceicne use in cancer and genetic outreach sterategies

Methods

Page 7, line 121, the authors should add the Protocol Approval Number from the Echics Committe

Page 8, line 137, Table 2 contains the results of data compelteness and should be included only in the results.

There is not a description of data statistical analysis process, eg using mean and SD

Results

In the academic writing, number less than 10 are suggested to be written in letters. The authors might conider writing these number is letters.

Discussion

The authors did not report an strenghts and limitation of their study.

The authors should describe the practical implication of their study.

Reviewer #2: Summary

The was an interesting paper, thank you for the chance to review it. Some small pick ups on consistent terminology will help improve this excellent paper.

Major issues: No major issues

Minor issues: The following points are mainly around consistency language, which if improved, will improve the readability of this paper.

Abstract:

‘nonprofessionals’ – should read non-professional researchers – as they may be professionals in other fields?

Data was/were line 38/40 – consistent?

Line 31 – 40 was complex and could be summarised more clearly or succinctly, for example, adding a heading ‘method’ and ‘results’.

Intro:

Line 54: the word citizen appears here for the first time and hasn’t been defined previously. Suggest clarification of terminology in first paragraph when defining patient and public involvement.

59: PPI collaboration or just PPI, if introducing the term collaboration does this need to be defined? Same with the word engagement.

61: ‘higher order citizen engagement’ – again, does this mean PPI? Consider using a task based approach rather than levels to describe involvement – as in ‘involvement in research design’ (high level) or just participation in survey responses (low level). A lot of this terminology will be unfamiliar to some readers so it’s important to be consistent in the terminology you are using.

63: ‘applied citizen science’ I would contest that you can ‘apply’ citizen science as a method, you could say ‘we involved people using methods informed by citizen science’ or words to that effect, if you see the distinction. At the very least you could say ‘we applied citizen science methodology’ but you’d need to carefully define what exactly you mean by that before using the shorthand later.

66: ‘Citizen science (CS) approaches involve nonprofessionals (citizens)’ suggest rewording as ‘Citizen science (CS) methods [metholodiges/paradigms?] involve non-professional researchers (citizens)

67: shouldn’t you add ‘collect data’ here too, as that is what happened with this project?

72: again, consider the word ‘approaches’ and defining it sooner. Methods, modes and tasks had very clear definitions in the review you are citing and I think this paper would be improved if terms like ‘citizen science approaches’ or ‘citizen science methods’ we used more consistently. I acknowledge this a wider issue of defining woolly terms such as CS (see my later comment about STARDIT), and you could consider using ‘participatory action research’ as an alternative term to CS.

75: regarding the line ‘Nunn and colleagues concluded that genomics- and related translation researchers could benefit from standardized methods to apply PPI’ you might be interested to learn about the project Standardised Data on Initiatives (STARDIT) which followed based on the recommendations of this scoping review and is now in a Beta stage of development. It allows stanardised reporting of involvement and the impacts of involvement (among much else). Pre-print here: https://doi.org/10.31219/osf.io/w5xj6 - and a friendly invitation to all authors to get involved with the STARDIT project if it’s of interest

82: do not advise the use of the word racial in the line ‘racial/ethnic minorities’, as race is generally considered a social construction. Ethnic acceptable as a short hand (if defined - .e.g is it cultural or are you specifically referring to inherited characteristics – e.g. groupings of people more pre-disposed to certain cancers with more awareness in those associated communities?) but perhaps also noting language communities and other cultural influences which might affect uptake. The point is you are grouping people by race in this comment, and people may not self-identify with such groupings. I note the papers you cite use this term, but in a health context I’d avoid it unless you are specifically talking about people’s lived experience of racism, which is distinct from reinforcing socially constructed racial categories. This article explains better what I’m trying to say: https://en.wikipedia.org/wiki/Race_(human_categorization)

85: This is a fantastic point and I note you’re now also introducing ‘co-create’. Again I suggest you chose a term at the start, mention all the different things it can be called, and then use that consistently as co-creation as a method might also need more explanation?

92: ‘Survivors’ is this a term that was co-defined by the people you involved, do they self-identify with that? Perhaps ‘people affected by cancer….’ Which also then includes loved-ones/carers of people with cancer etc

94: ‘Additionally, the credibility of researchers who share their passion for expanding reach could engage CS members more deeply in seeing the value of research.’ I think I know what you’re trying to say here, and I think it’s a good point. Could it be made into two sentences and made a little more explicit, perhaps with a ‘for example’? I think it’s an important point and worth expanding on.

96: ‘Individuals who learn they are at high risk for carrying inherited genetic mutations such as BRCA1/2 have the responsibility of (i.e., “duty to warn”) informing first- and second-degree blood relatives that they too may be mutation carriers’ – I would qualify this with a ‘Some individuals….may feel’ – otherwise you are prescribing a moral and ethical stance, which is dangerous waters. I know for example many people who feel it is not their duty and this might be a huge breach of ethics – unless they had asked if someone wanted to know such things first. Naturally this is a very complex area so you may wish to consult a qualified genetic councillor on writing about this accurately. If you wanted to take a CS/PAR approach to this, you’d actually say ‘CS can help understand different people’s attitudes towards sharing information about risk with relatives, and co-create learning and development interventions to support people to make informed decisions when communicating about such risk’ – or words to that effect if you see where I’m coming from?

103: typo – in sum.

107: ‘We based the procedures for the approach on implementation quality criteria

108 described by Heigl and colleagues’ – seeing as this is the basis for a large and complex table, a line or two about what this criteria is and how you applied it to this work might help the reader make more sense of the table (I had to go back and forth a couple of times and read the original paper before it clicked for me – but I might be slow!). line 175 explains it in more detail, and parts of this section could be moved earlier?

Methods and other sections

124: ‘Survivors’ – again, this term is used – is it distinct from other people involved as non-professional researchers (referred to on line 154 as ‘CS members’. Suggest this term is defined, or changed to ‘people affected by cancer (which can include people with, survivors of, people pre-disposed to, family of…etc – define as you wish). Later the acronym OCS is also used – being consistent as early as possible will help the reader navigate these complex terms.

125: the para FITD might need further review by someone more familiar with this method – this short section felt like it might need to be integrated further? In other words, why am I reading about it? Because this method was used? Could a sentence like ‘our method was informed by the FITD….because….) before hand make the link clearer?

146: stylistically, starting a new section with ‘thus’ might need looking at. I think it’s fine after a set up clause, but this is a matter of style, so take or leave.

161: ‘snowball sampling’ – while I know what you mean here, but noting that some readers might not (aside from the fact that some citizen science projects might actually sample actual snow!) perhaps a reference to this method for more information, or a short explanation of what that means?

218: ‘All CS members were female’ – does this paper require some commentary on this? Perhaps a line in the discussion around 284?

225: ‘listservs’ – define this term

252: does this line and others need putting in inverted commas ‘gaining insights about the experiences of ovariancancer survivors that will be useful to me,’?

286: ‘using probing techniques’ suggest you define this

6. PLOS authors have the option to publish the peer review history of their article (what does this mean?). If published, this will include your full peer review and any attached files.

Reviewer #1: **Yes: **Ilda Hoxhaj

Reviewer #2: **Yes: **Jack S Nunn

---

## [Author Response · Author response to Decision Letter 0]

18 Oct 2021

Review Comments to the Author

Reviewer #1

Introduction

Page 3 line 69, there is no reference for the sentence

“In the initial 63 planning year of a four-year NCI-funded study entitled “Your Family Connects” we applied citizen 64 science to generate intervention content for a message-based outreach intervention to encourage 65 families affected by ovarian cancer to consider genetic service options.”

References are noted in the version that was submitted at the line numbers suggested.

Page 4, line 75. There is no reference for the sentence “Nunn and colleagues concluded that genomics- and 75 related translation researchers could benefit from standardized methods to apply PPI”. This regards the ref 16, which should be reported also in this sentence, otherwise it seems that is the continuum of ref 17.

The reference number is now added.

Page 4, line 77-78, the sentence “Up to 20% of ovarian 78 cancers are linked to inherited genetic mutations in cancer predisposing genes”, should not be underlined.

The underline has been removed.

Page 4, line 89 It is not clear the transition to concerns, i would suggest to add a sentence on the general concerns before describing each of them in details.

A transition is now added.

Page 5, line 107-108, i would suggest to include Table 1 in the Methods section and leave in the introduction the reference.

Table 1 has been moved the “Methods” section.

The introduction is quite long and should be shortened, with a focus on Citizens sceicne use in cancer and genetic outreach sterategies

The introduction has been shortened by just over half a page.

Methods

Page 7, line 121, the authors should add the Protocol Approval Number from the Echics Committe

Protocol number has been added.

Page 8, line 137, Table 2 contains the results of data compelteness and should be included only in the results.

Table 2 has been moved to the “Results” section.

There is not a description of data statistical analysis process, eg using mean and SD

A few sentences are now provided to indicate that we used simple descriptive statistics.

Results

In the academic writing, number less than 10 are suggested to be written in letters. The authors might conider writing these number is letters.

Numbers under 10 (unless a Task descriptor) have been spelled out.

Discussion

The authors did not report an strenghts and limitation of their study.

The authors should describe the practical implication of their study.

The narrative now makes it clearer what narrative was intended to describe strengths and limitations.

Reviewer #2: Summary

Minor issues: The following points are mainly around consistency language, which if improved, will improve the readability of this paper.

Abstract:‘nonprofessionals’ – should read non-professional researchers – as they may be professionals in other fields?

Data was/were line 38/40 – consistent?

Line 31 – 40 was complex and could be summarised more clearly or succinctly, for example, adding a heading ‘method’ and ‘results’.

All of the above edits have been made to maintain consistency in terms and break up complex sentences.

Intro:

Line 54: the word citizen appears here for the first time and hasn’t been defined previously. Suggest clarification of terminology in first paragraph when defining patient and public involvement.

Terminology is now clarified up front.

59: PPI collaboration or just PPI, if introducing the term collaboration does this need to be defined? Same with the word engagement.

Now defined.

61: ‘higher order citizen engagement’ – again, does this mean PPI? Consider using a task based approach rather than levels to describe involvement – as in ‘involvement in research design’ (high level) or just participation in survey responses (low level). A lot of this terminology will be unfamiliar to some readers so it’s important to be consistent in the terminology you are using.

In shortening the introduction, this statement was omitted.

63: ‘applied citizen science’ I would contest that you can ‘apply’ citizen science as a method, you could say ‘we involved people using methods informed by citizen science’ or words to that effect, if you see the distinction. At the very least you could say ‘we applied citizen science methodology’ but you’d need to carefully define what exactly you mean by that before using the shorthand later.

Agree, we now describe as “engage CSs”.

66: ‘Citizen science (CS) approaches involve nonprofessionals (citizens)’ suggest rewording as ‘Citizen science (CS) methods [metholodiges/paradigms?] involve non-professional researchers (citizens)

Change has been made as suggested.

67: shouldn’t you add ‘collect data’ here too, as that is what happened with this project?

Collected was added.

72: again, consider the word ‘approaches’ and defining it sooner. Methods, modes and tasks had very clear definitions in the review you are citing and I think this paper would be improved if terms like ‘citizen science approaches’ or ‘citizen science methods’ we used more consistently. I acknowledge this a wider issue of defining woolly terms such as CS (see my later comment about STARDIT), and you could consider using ‘participatory action research’ as an alternative term to CS.

We now use terms consistently and use the terminology we used in the consent form and all CS activities. 

75: regarding the line ‘Nunn and colleagues concluded that genomics- and related translation researchers could benefit from standardized methods to apply PPI’ you might be interested to learn about the project Standardised Data on Initiatives (STARDIT) which followed based on the recommendations of this scoping review and is now in a Beta stage of development. It allows stanardised reporting of involvement and the impacts of involvement (among much else). Pre-print here: https://doi.org/10.31219/osf.io/w5xj6 - and a friendly invitation to all authors to get involved with the STARDIT project if it’s of interest

Thank you for making us aware of this resource. We were not able to figure out how to cite this effort in the narrative.

82: do not advise the use of the word racial in the line ‘racial/ethnic minorities’, as race is generally considered a social construction. Ethnic acceptable as a short hand (if defined - .e.g is it cultural or are you specifically referring to inherited characteristics – e.g. groupings of people more pre-disposed to certain cancers with more awareness in those associated communities?) but perhaps also noting language communities and other cultural influences which might affect uptake. The point is you are grouping people by race in this comment, and people may not self-identify with such groupings. I note the papers you cite use this term, but in a health context I’d avoid it unless you are specifically talking about people’s lived experience of racism, which is distinct from reinforcing socially constructed racial categories. This article explains better what I’m trying to say: https://en.wikipedia.org/wiki/Race_(human_categorization)

In shortening the manuscript, this sentence has been omitted.

85: This is a fantastic point and I note you’re now also introducing ‘co-create’. Again I suggest you chose a term at the start, mention all the different things it can be called, and then use that consistently as co-creation as a method might also need more explanation?

 Terminology has been made consistent.

92: ‘Survivors’ is this a term that was co-defined by the people you involved, do they self-identify with that? Perhaps ‘people affected by cancer….’ Which also then includes loved-ones/carers of people with cancer etc

Use of this term in now justified in the narrative.

94: ‘Additionally, the credibility of researchers who share their passion for expanding reach could engage CS members more deeply in seeing the value of research.’ I think I know what you’re trying to say here, and I think it’s a good point. Could it be made into two sentences and made a little more explicit, perhaps with a ‘for example’? I think it’s an important point and worth expanding on.

This point is now made in two sentences as suggested.

96: ‘Individuals who learn they are at high risk for carrying inherited genetic mutations such as BRCA1/2 have the responsibility of (i.e., “duty to warn”) informing first- and second-degree blood relatives that they too may be mutation carriers’ – I would qualify this with a ‘Some individuals….may feel’ – otherwise you are prescribing a moral and ethical stance, which is dangerous waters. I know for example many people who feel it is not their duty and this might be a huge breach of ethics – unless they had asked if someone wanted to know such things first. Naturally this is a very complex area so you may wish to consult a qualified genetic councillor on writing about this accurately. If you wanted to take a CS/PAR approach to this, you’d actually say ‘CS can help understand different people’s attitudes towards sharing information about risk with relatives, and co-create learning and development interventions to support people to make informed decisions when communicating about such risk’ – or words to that effect if you see where I’m coming from?

As written the sentence was not making the intended point that those found to carry mutations are by clinical and ethical standards the only ones to communicate with their families. We have now reworded the sentence to make this point clearer.

103: typo – in sum.

In shortening the intro, this sentence was omitted.

107: ‘We based the procedures for the approach on implementation quality criteria

108 described by Heigl and colleagues’ – seeing as this is the basis for a large and complex table, a line or two about what this criteria is and how you applied it to this work might help the reader make more sense of the table (I had to go back and forth a couple of times and read the original paper before it clicked for me – but I might be slow!). line 175 explains it in more detail, and parts of this section could be moved earlier?

Table 1 is now moved to the “Methods” section where it can be given more detailed description.

Methods and other sections

124: ‘Survivors’ – again, this term is used – is it distinct from other people involved as non-professional researchers (referred to on line 154 as ‘CS members’. Suggest this term is defined, or changed to ‘people affected by cancer (which can include people with, survivors of, people pre-disposed to, family of…etc – define as you wish). Later the acronym OCS is also used – being consistent as early as possible will help the reader navigate these complex terms.

See above response.

125: the para FITD might need further review by someone more familiar with this method – this short section felt like it might need to be integrated further? In other words, why am I reading about it? Because this method was used? Could a sentence like ‘our method was informed by the FITD….because….) before hand make the link clearer?

As written, this section was not make the point clearly that CSs were implementing tasks to develop a FITD guided intervention. We were not using the FITD approach on the CSs. These few sentences have been edited to make this point clearer.

146: stylistically, starting a new section with ‘thus’ might need looking at. I think it’s fine after a set up clause, but this is a matter of style, so take or leave.

Omitted

161: ‘snowball sampling’ – while I know what you mean here, but noting that some readers might not (aside from the fact that some citizen science projects might actually sample actual snow!) perhaps a reference to this method for more information, or a short explanation of what that means?

Has been defined and a reference has been added.

218: ‘All CS members were female’ – does this paper require some commentary on this? Perhaps a line in the discussion around 284?

Now described as a limitation

225: ‘listservs’ – define this term

Now defined.

252: does this line and others need putting in inverted commas ‘gaining insights about the experiences of ovarian cancer survivors that will be useful to me,’?

Now added.

286: ‘using probing techniques’ suggest you define this

Now defined.

---

## [Decision Letter · Decision Letter 1]

30 Dec 2021

Applying Citizen Science to engage families affected by ovarian cancer in developing genetic service outreach strategies

PONE-D-21-22428R1

Dear Dr. McBride,

We’re pleased to inform you that your manuscript has been judged scientifically suitable for publication and will be formally accepted for publication once it meets all outstanding technical requirements.

Kind regards,

Manuel Corpas, PhD

Academic Editor

PLOS ONE

Additional Editor Comments (optional):

Reviewers' comments:

Reviewer's Responses to Questions

**Comments to the Author**

1. If the authors have adequately addressed your comments raised in a previous round of review and you feel that this manuscript is now acceptable for publication, you may indicate that here to bypass the “Comments to the Author” section, enter your conflict of interest statement in the “Confidential to Editor” section, and submit your "Accept" recommendation.

Reviewer #2: All comments have been addressed

2. Is the manuscript technically sound, and do the data support the conclusions?

Reviewer #2: Yes

3. Has the statistical analysis been performed appropriately and rigorously? 

Reviewer #2: N/A

4. Have the authors made all data underlying the findings in their manuscript fully available?

Reviewer #2: Yes

5. Is the manuscript presented in an intelligible fashion and written in standard English?

Reviewer #2: Yes

6. Review Comments to the Author

Reviewer #2: Thank you for the opportunity to read such an interesting manuscript , I wish you all the best and hope we can stay in touch about http://scienceforall.world/STARDIT

7. PLOS authors have the option to publish the peer review history of their article (what does this mean?). If published, this will include your full peer review and any attached files.

Reviewer #2: **Yes: **Jack S Nunn

---

## [Editor Report · Acceptance letter]

4 Feb 2022

PONE-D-21-22428R1 

Applying Citizen Science to engage families affected by ovarian cancer in developing genetic service outreach strategies 

Dear Dr. McBride:

I'm pleased to inform you that your manuscript has been deemed suitable for publication in PLOS ONE. Congratulations! Your manuscript is now with our production department. 

Kind regards, 

on behalf of

Dr. Manuel Corpas 

Academic Editor

PLOS ONE